# Adverse Outcome in Non-severe COVID-19: Potential Diagnostic Coagulation Tests

**Rossella Cacciola [1,\*], Elio Gentilini Cacciola [2], Veronica Vecchio [3] and Emma Cacciola [4]**

[1] Hemostasis Unit, Department of Clinical and Experimental Medicine, University of Catania, 95123 Catania, Italy

[2] Policlinico "Umberto I", Department of Public Health and Infectious Diseases, "Sapienza" University of Rome, 00182 Rome, Italy; gentilini.elio@yahoo.it

[3] Hemostasis Unit, Medical School of Catania, University of Catania, 95123 Catania, Italy; vecchio.veronica99@gmail.com

[4] Hemostasis Unit, Department of Medical, Surgical Sciences and Advanced Technologies "G.F. Ingrassia", University of Catania, 95123 Catania, Italy; ecacciol@unict.it

**\*** Correspondence: rcacciol@unict.it; Tel./Fax: +39-(0)953781962

**Abstract:** COVID-19-associated coagulopathy (CAC) identifies the coagulation changes in coronavirus disease 2019 (COVID-19) and is responsible for thrombosis. CAC has been studied in critical and severe stage COVID-19 disease through tests including the D-Dimer (DD), prothrombin time (PT), thromboplastin partial time (PTT), platelet count, fibrinogen (Fib), and platelet factor 4 (PF4) tests. However, these tests have some limitations. The aim of this study was to identify more accurate warning tests for early recognition of CAC and to prevent its deterioration to disseminated intravascular coagulation (DIC). First, we measured Interleukin-1α (IL-1α) and IL-8, and tissue factor pathway inhibitor (TFPI) as inflammation and endothelial damage markers, respectively. Second, we measured thrombin antithrombin complex (TAT), β-Thromboglobulin (β-TG), and thromboelastometric parameters including clotting time (CT), clot formation time (CFT), clot firmness (MCF), and clot lysis at 30 min (LY-30), as markers of coagulation and platelet activation. This study included 100 non-severe patients with COVID-19 that developed pulmonary embolism (PE) compared to 80 healthy patients. IL-1α and IL-8, and TFPI were higher as well as TAT and β-TG and thromboelastometric parameters, indicating hypercoagulability. If confirmed in other studies, these results could help in predicting the deterioration of non-severe COVID-19 disease, thereby reducing hospitalizations and health costs.

**Keywords:** COVID-19; coagulation; platelet; thromboelastometry

## 1. Introduction

COVID-19 is a disease caused by a new coronavirus called severe acute respiratory syndrome (SARS) coronavirus 2 (CoV-2) [1]. Several studies defined COVID-19 as a highly thrombotic disease and showed that thrombosis affects morbidity and mortality [2–4]. COVID-19 thrombosis is linked to coagulation changes [5]. CoV-2 is not responsible for coagulation changes itself [6]. In fact, the inflammation seems to be responsible for coagulation changes [7]. A literature review documented deep inflammation associated with coagulation changes in severe and critical COVID-19 disease focusing attention on inflammation, endothelium, and coagulation assays. In this setting, it has been showed that IL-6 activates endothelium causing a endothelial dysfunction, characterized by infiltration of inflammatory cells and apoptosis of endothelial cells, known with the synonyms of "Endotheliopathy", [6] "Endothelialitis" [8], or "Endotheliitis" [9], which results in coagulation changes characterized by increase of tissue factor (TF) and activation of coagulation cascade, increase of Fibrinogen (Fib), which is a platelet-binding factor,

and increased thrombin which is also a platelet-activating factor [10,11]. D-dimer (DD), prothrombin time (PT), activated partial thromboplastin time (APTT), Fib, PF4, and platelet count are the most performed coagulation assays in COVID-19 studies [12]. However, each of these assays has some limitations. DD has low specificity and elevated levels are related with other conditions (advanced age, female sex, pregnancy, surgery, immobility, active malignancy, connective tissue disorders, end-stage renal disease, cocaine use, prior thromboembolic disease, and African American race) or reflect a later phase in the coagulation process since it is the final product of fibrin degradation induced by the fibrinolytic system. [13] Standard coagulation laboratory assays such as PT and APTT are a measure of the plasma clotting activity that ignore other components of the coagulation such as the platelets and the fibrinolysis [14]. The fibrinogen concentration is a static measure and does not provide information about its functionality. It is known that fibrinogen is an acute reactive protein which increases in earlier stage of disease indicating inflammation and decreases in later stage of disease, indicating consumption coagulation [14].

β-TG and PF4 are two platelet-secreted proteins that detect the increase of the platelet activation in vivo [15]. However, the plasma levels of PF4 are affected by a rapid remove from circulation through the binding to endothelial cells or by the mobilization from vascular endothelium on heparin infusion [16,17]. β-TG levels are not affected by these interference factors and therefore normal β-TG plasma levels exclude platelet activation in vivo while high levels indicate increased platelet activation in vivo. [15] Platelet count has some limitations as it does not provide information regarding the platelet functionality [18,19] and its increase may be the result of excessive inflammation; its decrease might be the result of immunological platelet destruction, impaired megakaryocitopoiesis, or consumption that reflects a later phase in the coagulation process [19]. From the background of the wealth the literature about assays which were performed, what remains to be studied are parameters indicating the severity of coagulation and deterioration of the disease. The aim of the present study was to identify more accurate diagnostic coagulation tests in order to facilitate early recognition of CAC and to prevent its deterioration.

## 2. Material and Methods

We designed a case-control study on hospitalized 100 patients (45 ± 5 years, range 30–45) with non-severe COVID-19 disease diagnosed according to WHO interim guidance [1] confirmed by a positive SARS-CoV-2 reverse transcriptase polymerase chain reaction test by nasopharyngeal/oropharyngeal swab and 80 blood donors (40 ± 5 years, 30–45) age-and sex-matched as controls. At admission, the COVID-19 disease patients had fever and respiratory symptoms without viral pneumonia on computed tomography scan (CT). AllCOVID-19 patients received empirical antimicrobial treatment such as amoxifloxacin and/or cephalosporin and antiviral therapy such as oseltamivir and/or ganciclovir. After a follow-up of 1 month, all 100 COVID-19 patients had worsening symptoms, showing the presence of pulmonary embolism (PE) on computed tomography (CT) angiography. None of the 100 patients went onto DVT documented by compression ultasonography. None of the 100 patients died, ended up the ICU on ventilators, ECMO, etc. In order to avoid confounding elements we chose subjects without comorbidities such as obesity, hypertension, diabetes, cancer, and heart disease (Table 1), for all patients and controls we measured the more commonly investigated coagulation/hematology or inflammation parameters in the laboratory of Hemostasis/Hematology Unit. Each study participant gave written informed consent for study enrollment in accordance with the Declaration of Helsinki. Blood collection was carried out at the admission and plasma preparation was obtained by centrifuging of anticoagulant citrate dextrose-anticoagulated whole blood at 2000 × *g* for 15 min. At the timing of blood collection none were on anticoagulant or corticosteroid therapy. Each laboratory marker was antigen and TFPI was free.

**Table 1.** Characteristics of coronavirus disease 2019 (COVID-19) patients.

| Anagraphic and Clinical Data | Patients, $n = 100$ | Controls, $n = 80$ |
|---|---|---|
| Age (yrs) | 45 ± 5 | 40 ± 5 |
| Male $n$ | 45 | 40 |
| Female, $n$ | 55 | 40 |
| **Respiratory support** | | |
| Oxygen supplementation | 0 | 0 |
| Mechanical ventilation | 0 | 0 |
| **Comorbidities** | | |
| Obesity | 0 | 0 |
| Hypertension | 0 | 0 |
| Diabetes | 0 | 0 |
| Cancer | 0 | 0 |
| Heart disease * | 0 | 0 |
| **Presenting symptoms** | | |
| Cough | 0 | 0 |
| Fever | 100 | 0 |
| Dyspnea | 100 | 0 |
| Headache | 0 | 0 |
| Anosmia | 0 | 0 |
| Pulmonary Embolism | 100 | 0 |

* Coronary artery disease or congestive heart failure.

*2.1. Laboratory Measurements*

2.1.1. Cytokines

To investigate the inflammation, we measuredInterleukin-1α (IL-1α) and IL-8 using ELISA kits (R&D Systems, Minneapolis, MN, USA) and analyzed them on a Luminex machine.

2.1.2. Endothelium

To investigate the endothelial damage, we measured TFPI using ELISA kit (American Diagnostic Inc., Stamford, CT, USA).

2.1.3. Coagulation and Platelet Activation

To investigate coagulation and platelet activation, we measured thrombin antithrombin complex (TAT) and β-TGusing ELISA kit (Diagnostic Stago, Boehringer Mannheim, Mannheim, Germany and R&D Systems, respectively). As D-dimer has been the biomarker most studied to date with COVID, the D-dimer levels were included in this study using the enzyme-linked immunosorbent (Diagnostic Stago, Boehringer Mannheim, Mannheim, Germany).

2.1.4. Blood Viscoelastic Analysis

To investigate the blood viscoelastic properties, we used the thromboelastometry method (Rotem delta System – Pentapharm, Wurzburg, Germany).

### 2.1.5. Statistical Analysis

The data set assumed the assumptions for the Student *t* test. The Pearson test and Spearman test were used for the correlations. The correlation coefficients were used to quantify the strength of the linear relationship. The samples were collected on day of admission and none of the patients were on anticoagulation or corticosteroid medication. Significance for all descriptive analysis was set at *p* <0.05 (SPSS 21.0 for Windows (SPSS Inc.)

### 3. Results

The studied parameters showed increased levels of IL-1$\alpha$ and IL-8 (2.9 ± 1 pg/mL vs 0.20±0.5 pg/mL and 60 ± 10 pg/mL vs 21 ± 3 pg/mL, respectively) as well as increased levels of TFPI (166 ± 69 ng/mL vs 81 ± 12 ng/mL) (Table 2). TAT and $\beta$-TG were increased in patients with COVID-19 (70 ± 10 μg/L and 245 ± 15 IU/mL) compared to controls (3 ± 1μg/L and 10 ± 5 IU/mL) (Table 2) as well as D-dimer (420 ± 100 μg/L) compared to controls (90 ± 80 μg/L). ROTEM analysis showed shortened CT (CT, unit: s. n.v. 100–240 s) (INTEM 45 ± 20 s, EXTEM 20 ± 10 s) and shortened CFT (CFT, unit: s, n.v. 30–160 s) (INTEM 15 ± 10 s, EXTEM 21 ± 10 s), increased MCF (MCF, unit: mm, n.v. 50–72 mm) (INTEM 120 ± 10 mm, EXTEM 115 ± 10 mm), and lower LY-30 (LY-30, %: v.n. 15%) (INTEM 0.8%, EXTEM 0.7%) in patients with COVID-19 compared to controls (INTEM CT 100 ± 10 s and CFT40 ± 5 s and MCF 70 ± 10 mm, and LY-30 15%, EXTEM CT 40 ± 10 s and CFT 60 ± 10 s and MCF 55 ± 5, and LY-30 15%) (Table 3). Positive correlations IL-1$\alpha$/TFPI (*r* = 0.862 rs = 0.890), IL-8/TFPI (*r* = 0.882 rs = 0.892), TFPI/TAT (*r* = 0. 872 rs = 0.869), and TAT/$\beta$-TG (*r* = 0.854 rs = 0.835) were found.

**Table 2.** Laboratory characteristics of coronavirus disease 2019 (COVID-19) patients.

| Biomarkers | Patients, *n* = 100 | Controls, *n* = 80 |
|:---:|:---:|:---:|
| IL-1$\alpha$, pg/mL | 2.9 ± 1 | 0.20 ± 0.5 |
| IL-8 pg/mL | 60 ± 10 | 21 ± 3 |
| TFPI, ng/mL | 166 ± 69 | 81 ± 12 |
| TAT, μg/L | 70 ± 10 | 3 ± 1 |
| $\beta$TG, IU/mL | 245 ± 15 | 10 ± 5 |

*p* values: < 0.05 compared with controls. Reference values of IL-1$\alpha$ (0.15–0.36 pg/mL), IL-8 (< 31.2 pg/mL), TFPI (75–120 ng/mL), TAT (1.0–4.1 μg/L), $\beta$TG (10–40 IU/mL).

**Table 3.** ROTEM parameters of coronavirus disease 2019 (COVID-19) patients.

| Thromboelastometry | Patients, *n* = 100 | Controls, *n* = 80 |
|:---:|:---:|:---:|
| INTEM | | |
| CT, s | 45 ± 20 | 100 ± 10 |
| CFT, s | 15 ± 10 | 40 ± 5 |
| MCF, mm | 120 ± 10 | 60 ± 10 |

| | | |
|---|---|---|
| LY, % | 0.8 | 15 |
| EXTEM | | |
| CT, s | 20 ± 10 | 40 ± 10 |
| CFT, s | 21 ± 10 | 60 ± 10 |
| MCF, mm | 115 ± 10 | 55 ± 5 |
| LY, % | 0.7 | 15 |

$p$ values: <0.05 compared with controls. INTEM test: reference values of CT (100–240 s), CFT (30–160 s), MCF (50–72 mm), LY (15%). EXTEM test: reference values of CT (38–79 s), CFT (34–159 s), MCF (50–72 mm), LY (15%).

## 4. Discussion

A number of immunological values were collected, and some were probably harbingers of hypercoagulability. O'Donnell J et al. [20] showed a correlation between pro-inflammatory cytokines (e.g., TNF-$\alpha$ and IL-1$\beta$) and expression of tissue factor on endothelial cells. We showed a correlation between pro-inflammatory cytokines (e.g., IL-1 and IL-8) and expression of TFPI in plasma. Schechter et al. [21] showed a correlation between pro-inflammatory cytokines (e.g., IL-1$\beta$ and IL-6) and thrombin contributing to tissue factor expression in monocytes. We showed a correlation between pro-inflammatory cytokines (e.g., IL-1 and IL-8) and TFPI contributing to TAT expression in plasma. Huang C. et al. [1] observed higher plasma levels of pro-inflammatory cytokines (e.g., IL-10 and IL-2) in Intensive Care Unit (ICU) patients. We observed higher plasma levels (e.g., IL-1 and IL-8) in PE patients. Concerning the results reported in present study, there are published studies reporting TFPI, TAT, or ROTEM results in COVID-19 patients. Particularly concerning TFPI and TAT levels, we obtained a mean level of 166 ng/mL and 70 ng/mL, respectively, in COVID-19 patients with pulmonary thrombotic disease, whereas White et al. [22] reported lower levels of TFPI and normal TAT in COVID-19 patients without thrombotic disease. Importantly, the finding of a particularly elevated TAT, a biomarker of thrombin generation, in our cohort of patients could be the result of severity of disease, assuming that the disease deteriorated into pulmonarythrombosis. This is consistent with the report by Jin et. al. [23] that reported higher TAT in COVID-19 patients and thrombosis. Regarding TFPI, it has been reported that plasma levels are affected by heparin which induces endothelial TFPI release causing artefactual high levels. In fact, White et al [22] studied critical patients treated with heparin and increased levels of TFPI which can reflect the contribution of heparin. Interestingly, we observed actual increased levels of TFPI in non-critical and non-heparinized patients. The sense of this datum could be because the elevated TFPI was on inflammated endothelium. ROTEM analysis is a point-of-care device that provides detailed information on clotting kinetics from the clot formation through degradation and is used to study the whole coagulation system [14]. Therefore, its use may be useful to predict the deterioration of COVID-19 disease towards thrombosis. There are many reports on TEG or ROTEM that look at COVID-19. Panigada et. al. [24] reported a hypercoagulable profile measured by TEG in 24 patients admitted to Intensive Care Unit (ICU) with COVID-19 showing shortened reaction time and clot formation time and a low LY-30. Similarly, Maatman et. al. [25], Wright et. al. [26]and Mortus et. al. [27] studied 12 and 44 and 21 patients admitted to ICU, respectively, and showed a hypercoagulable profile and fibrinolysis shutdown. Finally, Pavoni et. al. [28] conducted a retrospective study in ICU patients showing a hypercoagulability by ROTEM characterized by shortened CFT in INTEM and EXTEM, high MCF in INTEM and EXTEM and longer FIBTEM. However, these studiesfocus on the critical COVID-19 disease. Therefore, we studied non-ICU patients by ROTEM and found shortened CT in INTEM and EXTEM, shortened CFT in

INTEM and EXTEM, increased MCF in INTEM and EXTEM, and lower LY-30 in INTEM and EXTEM. The platelet activation is critical for thrombus formation. There is a wealth of literature on this topic that looks at measurement of plasma levels of PF4 and β-TG. A crucial question in the measurement of these proteins is the distinction between actual and artefactual in vivo levels. It has been reported that PF4 is a sensitive protein to heparin which mobilizes PF4 from binding sites on endothelial cells falsely increasing its in vivo plasma levels as reported in the articles by Busch [18] and Dawes [19]. Similarly, Hottz et. al. [29] and Middleton [30] reported increased plasma levels of PF4 in 35 and 14 heparinized ICU patients, respectively. In our study we found increased levels of β-TG and PF4 (unpublished data) in non-heparinized and non-ICU patients. If these findings may predict, the outcome deterioration in non-severe COVID-19 disease towards pulmonary thrombosis deserve to be confirmed and validated in larger studies.

**Author Contributions:** R.C., E.G.C., V.V. and E.C. equally contributed to the writing of this manuscript. All authors have read and agreed to the published version of the manuscript.

**Funding:** This research received no external funding.

**Institutional Review Board Statement:** The study was conducted in accordance with the Declaration of Helsinki and the protocol was approved by the Institutional Review Board "Hemostasis/Hematology Unit", University of Catania, (Q070/Q032), date of approval 20 January 2021.

**Informed Consent Statement:** All subjects gave their informed consent for inclusion before they participated in the study.

**Data Availability Statement:** Data will be available on request by email to rcacciol@unict.it.

**Conflicts of Interest:** The authors declare no conflict of interest.

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
