# Peer review of "Adverse Outcome in Non-Severe COVID-19: Potential Diagnostic Coagulation Tests"

_reports, doi:10.3390/reports4040035_

Round 1
Reviewer 1 Report
I have questions for the authors:
1. How many of the 100 patients with non-severe COVID went on to thrombosis and how many did not? Wouldn’t the patients who had non-severe COVID and did not go on to thrombosis be a better control group than the control group of age and sex-match blood donors in the current study?
2. How many of the patients went onto DVT and how many went onto PE, and how was this documented?
3. What was the ultimate outcome of the patients studied here? How many patients died, ended up in the ICU on ventilators, ECMO, etc.
4. As D-dimer has been the biomarker most studied to date with COVID, was there any attempt to include D-dimer levels in the study?
5. I do not understand the comments about TFPI levels and heparin in the Discussion. The authors found high levels of TFPI in the patients, and they state that this could be because the patients were not on heparin. However, if heparin elevates TFPI, how does this make sense – the high levels would be even higher, not lower, if the patients were on heparin.
Author Response
Revised Manuscript ID: Reports - 1373601
Type of manuscript: Article
point-by-point summary of responses
Title: Adverse outcome in non-severe COVID-19: potential diagnostic coagulation tests
Reviewer # 1
1.How many of the 100 patients with non-severe COVID went on to thrombosis and how many did not? Wouldn’t the patients who had non-severe COVID and did not go on to thrombosis be a better control group than the control group of age and sex-match blood donors in the current study?
Material and Methods
P3 – L10 The article “the” was deleted and the words “all 100” were added
2.How many of the patients went onto DVT and how many went onto PE, and how was this documented
Material and Methods
P3 – L12-13 The sentence was added as follows:
“None of the 100 patients went onto DVT documented by compression ultasonography”.
- What was the ultimate outcome of the patients studied here? How many patients died, ended in ICO on ventilators, ECMO, etc.
Material and Methods
P3 – L13-14 The sentence was added as follows:
“None of the 100 patients died, ended up the ICU on ventilators, ECMO, etc”.
4.As D-dimer has been the biomarker most studied to date with COVID, was there any attempt to include D-dimer levels in the study?
Laboratory measurements
P4 – L9-12 The sentence wsa added as follows:
“As D-dimer has been the biomarker most studied to date with COVID, the D-dimer levels were include inb this study using the enzyme-linked immunosorbent ((Diagnostic Stago, Boehringer Mannheim, Mannheim, Germany)”.
Results
P4 – L6-7 The sentence was added as follows:
“….as well as D-dimer (420±100 μg/l) compared to controls (90±80 μg/l).”.
5.I do not understand the comments about TFPI levels and heparin in the Discussion. The authors found high levels of TFPI in the patients, and they ssate that this colud be because the patients were not on heparin. However, if heparin elevates TFPI, how does this make sense – the high levels would be even higher, not lower, if the patients were on heparin.
Discussion
P5 - L7 The sentence was added as follows:
“….and non-heparinized patients”
P5 - L7 The sentence “…without contribution of heparin” was deleted.
P5 – L7-9 The sentence was added as follows:
“The sense of this datum could be because the elevated TFPI was on inflammated
Endothelium”.
Reviewer 2 Report
COVID-19-associated coagulopathy (CAC) identifies the coagulation changes in coronavirus disease 2019 (COVID-19) and it is responsible for thrombosis. The investigators aim to identify more accurate and warning tests for the early recognition of CAC and to prevent its deterioration to disseminated intravascular coagulation (DIC). A number of immunological values were collected and some (e.g., IL-10 and IL-8) were probably harbingers of hypercoagulability. The number of patients to support this claim may not be enough to support their conclusions and some limitations need to be expanded.
Author Response
Revised Manuscript ID: Reports - 1373601
Type of manuscript: Article
point-by-point summary of responses
Title: Adverse outcome in non-severe COVID-19: potential diagnostic coagulation tests
Reviewer # 2
1.Covid-19 associated coagulopathy (CAC) identifies the coagulation changes in coronavrus disease 2019 (COVID-19) and it is responsible for thrombosis. The investigators aim to identify more accurate and warning tests for the early recognition of CAC and to prevent its deterioration to disseminated intravascular coagulation (DIC). A number of immunological values were collected and some (e.g., IL-10 and IL-8) were probably harbingers of hypercoagulability. The number of patients to support this claim may non be enough to suport their conclusions and some limitations need to be expanded.
Discussion
P4-5 – L2-14 The sentences were added as follows:
“A number of immunological values were collected and some were probably harbingers of hypercoagulability. O’Donnell J et al. [20 ] showed a correlation between pro-inflammatory cytokines (e.g. TNF-α and IL-1β) and expression of tissue factor on endothelial cells. We showed a correlation between pro-inflammatory cytokines (e.g. IL-1 and IL-8) and expression of TFPI in plasma. Schechter et al. showed a correlation between pro-inflammatory cytokines (e.g. IL-1β and IL-6) and thrombin contributing to tissue factor expression in monocytes [21]. We showed a correlation between pro-inflammatory cytokines (e.g. IL-1 and IL-8) and TFPI contributing to TAT expression in plasma. Huang C et al. [22] observed higher plasma levels of pro-inflammatory cytokines (e.g. IL-10 and IL-2) in Intensive Care Unit (ICU) patients. We observed higher plasma levels (e.g. IL-1 and IL-8) in PE patients.
P4 - L3 The reference [20] was quoted
P4 – L7 The reference [21] was quoted
P5 – L11 The reference [22] was quoted
Round 2
Reviewer 1 Report
- I still have great difficulties with the discussion of this paper. The authors make a number of comments about their results being related to the patients being very sick or about to deteriorate, but on questioning, none of the patients went onto thrombosis or went on to need ICU care. Examples:
“ Particularly concerning TFPI and TAT levels, we obtained a mean level of 166 ng/ml and 70 ng/ml, respectively, in COVID-19 patients with thrombotic disease, whereas White et al [23]reported lower levels of TFPI and normal TAT in COVID-19 patients without thrombotic disease. Importantly, the finding of a particularly elevated TAT, a biomarker of thrombin generation, in our cohort of patients could be the result of severity of disease, assuming that the disease deteriorated into thrombosis, and consistent with the report by Jin et al [24]that reported higher TAT in COVID-19 patients and thrombosis”
“In our study we found increased levels of b-TG in nonheparinized and non-ICU patients. If these findings may predict the outcome deterioration in nonsevere COVID-19 disease towards thrombosis deserve to be confirmed and validated in larger studies.“
- The authors talk about high TAT levels in their controls because they are not heparinized. This statement does not make sense to me……the levels of TAT should be the baseline levels that a control patient would have – it has nothing to do with heparin being present or not present for these patients, since they are controls.
“ Another factor which may contribute to observed high TAT levels lies in our cohort of non-heparinized patients. Infact, there are studies on low or normal levels of TAT in heparinized patients. Paparella et al [25]reported that low TAT levels were associated with high heparin doses, Mombelli et al[26]observed twenty-four hours after heparin was discontinued a significant increment in TAT plasma levels, and Hofman et al [27]showed that TAT levels progressively normalized after heparinization”
- These two sentences need to be corrected for their English usage and grammar…..
“However, these studies make little sense because look at critical COVID-19 disease.”
“Similarly, Hottz et al [33] and Middleton[34] studied 35 and 14 ICU patients, respectively, and found increased plasma levels of PF4 which can make little sense because look at patients on heparin.”
Author Response
Revised Manuscript ID: Reports - 1373601
Type of manuscript: Article
point-by-point summary of responses
Title: Adverse outcome in non-severe COVID-19: potential diagnostic coagulation tests
Reviewer
1.I still have great difficuties with the discussion of this paper. The authors make animber of comments about their results being related to the patients being very sick or about to deteriorate, but on questionino, none of the patients went onto thrombosis or went on to need ICU care.
Discussion
P5 – L7
The word “….pulmonary…” was added.
P5 L-11
The word “….pulmonary…” was added.
P5 – L49
The sentence “….and PF4 (unpublished data)….” was added.
P6 – L1
The word “…. pulmonary…” was added.
2.The authors talk about high TAT levels in their controls because they are not heparinzed. This statement does not make sense to me….the levels of TAT should be the baseline levels that a control patients would have – it has nothing to do with heparin being present or not present for these patients, since they are controls.
P6 – L13-18
These three sentences:
“Another factor which may contribute to observed high TAT levels lies in our cohort of non-heparinized patients.”
“Infact, there are studies on low or normal levels of TAT in heparinized patients.”
“Paparella et al [25]reported that low TAT levels were associated with high heparin doses, Mombelli et al[26]observed twenty-four hours after heparin was discontinued a significant increment in TAT plasma levels, and Hofman et al [27]showed that TAT levels progressively normalized after heparinization.” were deleted.
3.These two sentences need to be corrected for their English usage and grammar….
P5 – L36
The word “Hovewer…” was corrected as follows:
“However….”.
P5 – L37
The sentence “….make little sense because look at…” was deleted and corrected as follows:
“…..focus on the….”.
P5 - L46-48
The sentence “….studied 35 and 14 ICU patients, respectively, and found increased plasma levels of PF4
which can make little sense because look at patients on heparin….” was deleted and corrected as follows:
reported increased plasma levels of PF4 in 35 and 14 heparinized ICU patients, respectively.
References
P9 L3-12
The references n. 25, n. 26 and n. 27 were deleted.